# Identification of People in a Household Using Ballistocardiography Signals Through Deep Learning

**DOI:** 10.3390/s25061732

**Published:** 2025-03-11

**Authors:** Karin Takahashi, Yoshinobu Tanno, Hitoshi Ueno

**Affiliations:** Faculty of Information Design, Tokyo Information Design Professional University, Edogawa-ku, Tokyo 132-0034, Japan; takahashi@tid.ac.jp (K.T.); tanno@tid.ac.jp (Y.T.)

**Keywords:** biological signal, personal identification, deep learning, monitoring system, piezoelectric sensors

## Abstract

Background: Various sensor technologies have been developed to monitor the health of older adults; however, most of them require attachment to the skin. This study aimed to develop a health monitoring system, using a non-adhesive, non-invasive polyvinylidene difluoride piezoelectric sensor, with the patient being able to lead a normal daily life without being conscious of the sensor. The vibration signal from the human body surface obtained by the piezoelectric sensor, which is a ballistocardiography signal, contains information on the person’s heart and respiratory rates. We propose a method that enables individual identification based on the characteristics of the frequency components of the signal. Methods: Signals from ten subjects were acquired and a neural network was constructed, trained, and tested using 252 cases to identify five individuals, based on assuming the number of people in a household. Results: The classification probability and accuracy rate were obtained for all 252 cases, and good classification rates were obtained in almost all cases. Conclusions: Although it will be necessary to consider daily changes in such signals in the future, the system had good identification accuracy when five individuals were identified.

## 1. Introduction

Recently, the need for health monitoring for the elderly within households has been increasing with the advancement of an aging society [1,2]. A technology for monitoring the health conditions of senior citizens living in their homes was proposed as a remote patient monitoring (RPM) system [3,4,5,6]. Shaik et al. reported the survey results on the physical activity monitoring technology in reviews [4]. Most of these technologies were non-invasive for patients; however, they require close alignment of the sensor with the skin.

Among elderly people who live their everyday life without recognizing themselves as patients, there are some who dislike making an effort to closely align a sensor with their skin to manage their health. Using sensors that are non-invasive without skin contact is necessary to monitor patient health conditions during daily life, including in elderly people. Using non-invasive methods without skin contact implies that elderly people can monitor their health status during everyday life without the need to wear sensors or be aware of them.

Strong candidates for sensors without skin contact are resistive film-type pressure sensors and polyvinylidene difluoride (PVDF) piezoelectric sensors. These sensors enable the detection of vibrations caused by biological signals, even through clothing. Among these, the PVDF piezoelectric sensor, which is a differential sensor, is appropriate for obtaining frequency responses from biological signals. Thus, we used a PVDF piezoelectric sensor to monitor the patient’s health status without skin contact.

Conventionally, the heart and respiratory rates gained from the vibration signal from the body surface are obtained using a PVDF piezoelectric sensor [7,8]. A diagram of these vibration signals is called a ballistocardiogram (BCG), and it is a signal originating from the patient’s heartbeat. For example, attaching this sensor to a bed allows BCG signals from the human body during sleep to be acquired and analyzed, which enables sleep states to be assessed, such as sleep depth [9]. However, this information is limited to heartbeat, respiratory, and body movement vibrations. The BCG signal comprises a frequency component that varies among individuals [10] and, therefore, individuals can be classified based on the features of the frequency component.

In previous studies on personal identification methods using biological signals, there are methods that use piezoelectric sensors, electroencephalograms [11,12,13], electrocardiograms [14,15], millimeter-wave/microwave radars [16,17], iris information [18], and fingerprint information [19]. Each method has its own characteristics. The acquisition of biological signals, such as brain waves and electrocardiograms, requires sensors that make direct contact with the skin. In this case, an analysis with a high level of accuracy is possible because the noise interference is small; however, the degree to which the monitored person is aware of the sensor must be considered. Even with methods that use iris or fingerprint information, the person being monitored must consciously move their eyes or fingers in regard to the sensor position and, therefore, these are not identification methods that occur naturally during daily life. The measurement method using millimeter waves and a microwave radar is a non-invasive measurement method for biological signals. However, care must be taken when handling radar-related measuring equipment. In contrast, the method that uses a piezoelectric sensor is easy to handle and enables non-invasive personal identification.

In this paper, we suggest a calculation method for possible personal identification from biosignals obtained using a PVDF piezoelectric sensor. The goal is to realize a personal authentication system that can identify individuals whose characteristics have been registered in advance, based on the characteristics of their frequency components. In this report, we show that individuals can be identified with reasonable accuracy, at least within the recognition range of five people, by using the individual differences in BCG signals obtained with a PVDF piezoelectric sensor. The accuracy of identifying five people was used as a benchmark, based on assuming the identification of the residents in a household. It is assumed that this sensor will be installed on chairs or beds in the home as a constant health monitor and can be used for identifying who is standing above the sensor. A neural network model was adopted as the calculation method to identify the individuals. Groups were created in which five people were extracted from the signal data of ten people, and a neural network model was constructed for solving a five-classification problem for all the cases. Training and testing were performed, and a good classification rate was obtained.

This paper describes the characteristics of the sensor used, compares it with other sensors, and explains the process related to the acquisition of biological signals. Further, we explain how to create the input data for the neural network from the acquired signal data, the calculation method, and the results from the neural network. Finally, we consider the effectiveness of the identification using these calculations.

## 2. Sensor Selection and Data Acquisition

### 2.1. Sensor Selection

Sheet-type pressure sensors that can detect body vibrations without the user being aware of them include piezoelectric sensors made of PVDF and pressure sensors made of thin-film resistors. Piezoelectric sensors are differential sensors that detect changes in pressure, while the latter are sensors that detect pressure directly.

The signals obtained when approximately the same pressure was applied to both the resistive film pressure sensor and piezoelectric sensor at the same time were compared. The experimental setup is illustrated in Figure 1. The resistive film pressure sensor and piezoelectric sensor were stacked one above the other, pressure was applied from above, and the waveforms were measured using an oscilloscope (Figure 1(1)). The resistive film pressure sensor requires a power source and, therefore, the potential voltage divided by a fixed resistor, R (1 KΩ), was measured using an oscilloscope (Figure 1(2)). A piezoelectric sensor generates a small amount of charge because of changes in pressure, creating a potential current; therefore, it is directly connected to the input of an oscilloscope, which has high input impedance for measurement purposes (Figure 1(3)). The resistive pressure sensor used was the FSR406 model from Interlink Electronics (Irvine, CA, USA) and the piezoelectric sensor was the sheet sensor “Shin-shin Lambda” from Health Sensing Co., Ltd. (Tokyo, Japan). The PVDF piezoelectric sensor was selected for use in this study due to its relatively high sensitivity compared to other piezoelectric sensors.

Figure 2 shows the measurement results when pressure was applied from above using this experimental setup. The signals from both sensors changed almost simultaneously, because pressure was applied from above onto the two stacked sensors. The upper signal waveform is the pressure measurement result obtained using a resistive membrane pressure sensor. The potential rises when pressure is applied and decreases when the pressure is removed. The lower signal waveform is the measurement result using a piezoelectric sensor, and the potential rises on the positive side when the moment pressure is applied and swings to the negative side when the moment pressure is removed.

Examining the change in the signal between time (A) and time (B) in Figure 2 indicates that the value of the piezoelectric sensor exhibits a shape that corresponds to the derivative of the signal from the resistive membrane pressure sensor. In this study, the frequency spectrum of BCG signals was used as judgment data for performing calculations for the personal identification. The sensor possesses the ability to generate a differential signal from the original signal, which implies that it is suitable for use in regard to this judgment method. The BCG signal generated by the human body originates from the BCG of the arteries, because blood pressure fluctuations are relatively gentle and high-frequency components decrease rapidly when spectral analysis is performed. In contrast, a differential-type sensor makes it possible to observe frequencies that are *n* times the fundamental frequency and *n* times the intensity. 

Any periodic function $f\left(x\right)$ can be expressed as a Fourier series:(1)$$f\left(x\right)=\frac{a_{0}}{2}+\sum_{n=1}^{\infty }\left(a_{n}\cos\left(nx\right)+b_{n}\sin\left(nx\right)\right),$$
where $a_{0}$ represents the DC component and $a_{n}$ and $b_{n}$ represent the Fourier coefficients, which represent the amplitudes of the cosine and sine terms, respectively. By taking the derivative of $f\left(x\right)$ with respect to x, the result is:(2)$$f^{\prime }\left(x\right)=\sum_{n=1}^{\infty }\left(-na_{n}\sin\left(nx\right)+nb_{n}\cos\left(nx\right)\right).$$

The term *n* times the period in $f^{\prime }\left(x\right)$ represents *n* times the signal strength compared to the term with the fundamental frequency of $f\left(x\right)$. This implies that the harmonic components are emphasized.

We chose to use it to measure body vibrations because of the advantageous characteristics of the piezoelectric sensor in regard to frequency-related spectral analysis.

### 2.2. Sample Dataset

In this study, the data were obtained from ten participants. The data include participants, namely two individuals in their 60 s, two in their 30 s, and six in their 20 s. The data acquisition experiment was conducted by placing a piezoelectric sensor sheet on a chair, and the participant sat on it (Figure 3). The signal generated by the piezoelectric sensor was acquired through a signal amplifier at a sampling frequency of 100 Hz, and the voltage values were transmitted to a personal computer (PC). The acquired signal band was in a low-frequency range around 50 Hz, allowing the use of a generic sensor data acquisition device. However, it is important to note that if the input impedance of the signal amplifier was low, the effective voltage was not measured. In this study, an operational amplifier with high input impedance was used, and both an amplifier and an analog/digital converter circuit were designed and connected to a data recording PC. Additionally, a data collection program was designed to run on the PC. The participant in the experiment waited quietly in a waiting room in the laboratory 5 min before the start of the measurement and, then, the participant sat in a chair that had the sensor installed when the experiment began. Since the objective was to obtain health monitoring data during normal daily life, the BCG signals were collected while the subjects were seated in a chair as usual. Body movement has a significant impact on sensor signals. However, in this study, we did not carry out continuous monitoring throughout the day, including during periods when the subjects made large movements. Instead, we planned for the BCG signals to be acquired and monitored multiple times throughout the day when the subjects were at rest. Therefore, the measurement duration was set to approximately three minutes. However, if prolonged signal disturbances occurred due to body movement, the acquisition time was extended. This was because data collected during resting periods were necessary to ensure classification accuracy when using a neural network (NN). The measurement method was identical to that used in a previous report [20].

In this study, the BCG signals are measured for ten subjects over the course of ~3 min. Figure 4 shows the obtained BCG signals for ten people, who are identified according to ID M0–M9. These signals were stable for seven people (ID M0–M6). The other three people (ID M7–M9) exhibited body movements; however, since the movements were during short periods, the measurement time was not significantly extended, and only about three minutes of data were used. In this study, we set the threshold for determining body movement as an amplitude fluctuation of approximately twice the average amplitude, including cases where the signal exceeds the scale limit. Out of the ~3 min of data, ~2 min of data were used as training data, and the remaining 1 min of data were used as test data. Figure 4 shows a typical spectrum that was extracted for 10 s and converted using fast Fourier transform (FFT).

### 2.3. Five-Classification Analysis

A five-person classification was used to analyze five-class problems using a neural network (NN). First, the ballistocardial signals for five people were selected from those of ten people. This combination pattern comprised 252 combinations. For each pattern, the NNs were trained, and the classification probabilities were obtained.

Figure 5 presents an overview of the training process for a NN in the case of a specific pattern (e.g., ID a, b, c, d, and e). During this process, the NN was trained using the training data for each of the IDs and the labels represented by 1 and 0. In regard to the training results, the classification probabilities were used as the output. The NN is trained such that each NN belongs to its own class in regard to the input of the five ID datasets.

Figure 6 indicates what happens when test data different from the training data are input into the pretrained NN, in regard to the pattern shown in Figure 5. The pretrained NN inputs new data independent of each ID and outputs the classification probabilities, which is the probability of each data point belonging to each class. Based on these results, we determined whether the test data were correctly classified and calculated the classification accuracy rate for each ID. The classification accuracy rate indicates the proportion of correct classifications. In this study, the classification accuracy rate ${\left(R_{Correct}\right)}_{k}$ for a specific ID k (where k⊆{a, b, c, d, e}, as shown in Figure 5, and k⊆{M0,M1,⋯M9} in this study) was calculated using:(3)$${\left(R_{Correct}\right)}_{k}=\frac{\sum_{i=1}^{m_{k}}n{\left(k\right)}_{i}}{m_{k}},$$
where $i\;(where\;i=1,2,\;\cdots ,\;m_{k})$ represents a specific test data, $m_{k}$ represents the total number of test data associated with the ID k. Further, $n{\left(k\right)}_{i}$ is 1 if classified correctly (where the probability of belonging to one’s own class $P{\left(k\right)}_{i}$ is larger than or equal to 0.5) and 0 if it is not classified correctly in regard to the test data i.(4)$$n{\left(k\right)}_{i}=\left\{\begin{array}{c} 1\;\;\;\;\;\;(P{\left(k\right)}_{i}\geq 0.5)\\ \;\;0\;\;\;\;\;\;(P{\left(k\right)}_{i}<0.5)\; \end{array}\right..$$

The NNs were trained, and the classification probabilities and accuracy rates for each ID were calculated for all 252 possible patterns.

#### 2.3.1. Creation of Training and Test Dataset

The data for ten individuals were divided into ~2 and 1 min segments. The former was used as training data, and the latter was used as test data. The training data were extracted in segments with a length of 1000 (equivalent to 10 s of measurement time), and the extraction time was randomly varied to create 500 sets for each individual. The test data were similarly extracted in segments with a length of 1000; however, the extraction time was sequentially shifted at 1 s intervals, which resulted in 50 sets for each individual. The extracted training and test data were transformed into frequency spectra using FFT. Low-frequency components near 0 Hz were replaced with zero and high-frequency components above 15 Hz (unrelated to the heart rate frequency components) were removed. The final dataset length was reduced to 150. Normalization was applied to each dataset separately, with the minimum value of each set mapped to zero and the maximum value mapped to one. The normalized values were calculated using:(5)$$x^{\prime }=\frac{x-min(X)}{\max\left(X\right)-min(X)},$$
where x represents an individual data point, min⁡(X) and max⁡(X) represent the minimum and maximum value of the entire dataset, respectively, and x′ represents the normalized value.

#### 2.3.2. Structure of the Neural Network

The deep learning model used was a simple feedforward NN model, with input, hidden, and output layers. The input data, with a length of 150, were fed into the NN as a 150 dimensional vector, $x\in R^{150}$. Then, the input data were passed to the input layer, which was a dense layer with 32 neurons. In regard to this layer, the input data were transformed linearly using a weight matrix $W^{\left(1\right)}$ and a bias vector $b^{\left(1\right)}$ and, therefore, a vector of $h^{\left(1\right)}\in R^{32}$ was achieved. The rectified linear unit (ReLU) activation function was applied element wise. The ReLU function returns a value if it is positive and zero if it is negative. The operation of the input layer is:(6)$$h^{\left(1\right)}=ReLU(W^{\left(1\right)}x+b^{\left(1\right)}).$$

The output of the input layer, $h^{\left(1\right)}\in R^{32}$, is passed to the hidden layer. This layer was dense with 16 neurons, and was followed by ReLU activation. The output is a vector, $h^{\left(2\right)}\in R^{16}$, which is passed to the next layer. The operation of this layer is:(7)$$h^{\left(2\right)}=ReLU(W^{\left(2\right)}h^{\left(1\right)}+b^{\left(2\right)}).$$

Finally, the output $h^{\left(2\right)}\in R^{16}$ was passed to the output layer, which was a dense layer with five neurons, corresponding to the five classes in regard to the classification. In regard to this layer, a linear transformation was applied, followed by softmax activation. The softmax function outputs a probability distribution, such that each output value represents the predicted probability of the input belonging to a particular class. The output was a vector, $\hat{y}\in R^{5}$, which contained the predicted probabilities for each class. The operation of the output layer is:(8)$$\hat{y}=Softmax(W^{\left(3\right)}h^{\left(2\right)}+b^{\left(3\right)}).$$

This NN was designed for a five-class classification task, and the cross-entropy loss function was:(9)$$L=-\frac{1}{N}\sum_{i=1}^{N}\sum_{k=1}^{5}y_{i,\;k}\log({\hat{y}}_{i,k})\;.$$
where N represents the number of samples; $y_{i,\;k}$ represents the true class label for sample i, and ${\hat{y}}_{i,k}$ represents the predicted probability of class k for sample i. The cross-entropy loss evaluates the gap between the predicted probabilities and the true class labels, encoded in a one-hot format.

The suitable number of datasets and epochs was selected by evaluating the accuracy rate of the test data across multiple datasets and epochs. The results were affected by a small number of sets, with 100 datasets for each individual. However, with more than 200 datasets, the effect of the limited number of sets on the results was reduced. For 500 sets, the results were almost independent of the number of datasets. The number of epochs was determined by monitoring the change in the loss value and ensuring that it was set within a certain range to prevent overfitting.

## 3. Results

Based on a dataset of 10 people, each ID from M0–M9 was selected and classified into five groups. In total, 252 different combinations of data were obtained.

### 3.1. Processing Time

In a single five-class classification, each class contained 500 datasets, with a total of 500 × 5 = 2500 datasets for all five classes combined. The training time required for a single five-class classification was ~10 ms. The inference processing time for 250 (each class containing 50) tested datasets was ~2 ms. Figure 7 shows the typical loss value for each epoch.

### 3.2. Five-Class Classification Results

The classification accuracy for all 252 possible combinations is shown in Figure 8. These results represent the classification accuracy when a dataset with five sets of labels was used for training and the trained model was then used to classify the test data. The classification accuracy for the five individuals is shown for each training instance. The classification accuracy is considered successful when the classification probability for the ID of the individual is 0.5 or higher. Then, the accuracy is represented as the percentage of correct predictions out of the 50 test data points for each individual. Of the 252 patterns or a total of 1260 classification results (252 × 5), only one instance had a classification accuracy below 0.5, which indicates that the classification was successful in almost all cases.

Among these, the top three (Figure 9), bottom three (Figure 10), and middle three results (Figure 11) are presented, with the classification probabilities displayed in each figure. Figure 9 shows that the classification probability indicates the probability of each individual being classified into one of five categories based on the results of the test dataset (50 sets). For one training run, the probability of each test data point belonging to a particular class was calculated, and the average probability of the classification results obtained from the 50 sets was represented as a bar graph. To clarify this, the probability of being classified in an individual’s own category is labeled as positive, whereas the probabilities of being classified into other categories are labeled as negative.

## 4. Discussion

### 4.1. Five-Class Classification

There was only one instance in which the accuracy was less than 50% in terms of the five-class classification results for the individual heart rate signals; this can be considered an excellent outcome. One case of low accuracy was attributed to significant body movements. This indicates that classifications with a high level of accuracy can be achieved by selectively choosing the data used for training the model. For the test data, many instances had an accuracy of 1; however, there were some cases in which the classification failed sporadically. However, in terms of an actual monitoring system for individual identification, observations are made over time, because the individual can still be identified by examining surrounding cases, even if a classification fails sporadically. Therefore, such instances were not problematic.

A detailed analysis of the optimal classification pattern was carried out, which was the “02367” pattern of IDs. Visualization techniques were employed to investigate which parts of the data the NN focuses on, and which features are important for the best classification pattern. A masking process was applied to specific peaks in order to eliminate their influence, followed by re-inference to examine changes in the classification probabilities. Masking refers to the replacement of numerical values with zero values for the masked frequency. Three types of masking were implemented: (1) masking from the low-frequency side; (2) masking from the high-frequency side; and (3) masking a single specific peak.

Figure 12 and Figure 13 illustrate the inference results after masking, selecting spectral ID M0 and ID M7 among the five subjects with the best classification (“02367” classification) scenario. For the classification of ID M0, a significant decrease in the classification probability was observed at ~2 Hz when masking was applied from the low-frequency side, indicating that the low-frequency peak was a major feature. Further, a decrease in the classification probability was observed when the peak at 7.5 Hz was removed, which suggests that the main peak was also recognized as a key feature, in addition to the low-frequency peak.

In contrast, for the classification of ID M7, the absence of low-frequency (approximately 0–4 Hz) or high-frequency (approximately 6–15 Hz) peaks did not significantly affect the classification probability. However, the absence of the main peak at 5 Hz resulted in a significant drop in the classification probability, which indicates that the shape of this peak was a key feature.

These observations lead to the conclusion that the classification probability is higher when the shape of the main peak is well-defined or when distinct peaks are present in both low- and high-frequency regions, thereby ultimately contributing to improved classification accuracy.

### 4.2. Features in Regard to the Worst Case

The worst cases in regard to the classification pattern, which was the “05689” case, were investigated to establish which parts of the data and which features the NN focuses on. Among the “05689” cases, the one with the worst classification probability was ID M9, which was often misclassified as ID 6. Therefore, the characteristics of ID M6 and M9 were investigated in this study.

Figure 14 and Figure 15 show the re-inference results after masking for the classifications in regard to the cases with the worst “05689” outcome, thereby focusing on ID M6 and ID M9. For ID M6, several distinct peaks were clearly visible in the frequency spectrum. Figure 14 shows that the significant loss of the main peak near 7.5 Hz notably reduced the classification probability, which indicates that the shape of this peak is a key feature.

In contrast, ID M9 lacked prominent peaks overall, indicating general ambiguity. Although the classification probability is low across the board, the loss of peaks near 5–7.5 Hz contributes further to a decrease in the classification probability. It is hypothesized that ID M9 is erroneously classified as ID M6 when the minor peak near 7.5 Hz becomes dominant because of the lack of distinguishing features.

However, ID M9 exhibited larger spectral variations depending on the measurement duration compared to the other ID, which led to a lower classification accuracy for ID M9 in general. The lack of prominent peaks was likely because of significant motion artifacts during measurement. By collecting BCG signals at rest for a long period of time and setting strict amplitude fluctuation thresholds for body movement conditions, the characteristics of the frequency spectrum are clarified. Therefore, it is possible to further improve the classification accuracy by selecting only measurement periods without motion artifacts, where frequency peaks are clearly extracted, and by carefully curating the spectra used as training data.

### 4.3. Limitations

Figure 16 shows the relationship between the dispersion of the training and test data and the classification accuracy rate. The dispersion of the training and test data is the standard deviation across all channels of the frequency spectra using the training or test data. Based on this relationship, dispersion did not significantly affect the classification accuracy rate. However, the dispersion for ID M9 is larger than that of the others, and the removal of spectrum data for a specific measurement time, which includes the effects of body movement, is considered to improve the classification accuracy.

Although the BCG spectrum for ID M5 included little body movement, the classification accuracy rate for ID M5 is lower than that of the others. The dispersion for ID M5 was only slightly larger. The shapes of the frequency spectra of ID M5 and M9 were similar. Therefore, there were more cases of incorrect classification. Assigning additional classified data, such as weights, is necessary to improve the classification accuracy rate for data with similar shapes, in addition to increasing the number of training data inputs into the neural network.

The current investigation involved training a NN in regard to BCG signals measured within the same timeframe, by segmenting the time domain and testing the BCG signals from different time segments, in order to demonstrate the classification accuracy rate. This implies that the frequency spectra of the BCG signals for each person had a different shape and, therefore, it is possible to classify each person. This study demonstrated that the five-person classification corresponded to the average number of people in a household. Training the NN considering time variations, such as the measurement date and time window, is necessary to ensure it is suitable for practical use. In addition, it will be necessary to train the NN using BCG signals from various postures, such as lying down or standing. These are the tasks for the next stage, for which we will likely need to collect a large amount of data.

## 5. Conclusions

A non-contact, non-invasive PVDF piezoelectric sensor was used to obtain BCG signals transmitted via the body surface, and individuals were identified based on the characteristics of the frequency components of these signals. A good classification accuracy rate was achieved for 252 combinations of the five categories of data from ten subjects. The main peak information was captured as a feature in the data, with the main peak achieved during the operation of the trained neural network investigated, and the presence of the main peak was found to improve the classification accuracy rate. In contrast, the classification accuracy rate tended to decrease in regard to frequency spectra without a main isolated peak. Further, it was confirmed that variations in the intensity of the frequency spectrum did not significantly affect the classification accuracy rate. These results confirm that spectral features can be extracted using a NN and that individuals from approximately one family can be identified from the characteristics of the frequency components of their BCG signal.

In future, we plan to build a more robust NN that considers daily changes and adds secondary information, such as weights, to improve the accuracy of individual identification. In addition, we plan to evaluate classification accuracy using more complex architectures, such as convolutional neural networks and time-series processing methods.

## Figures and Tables

**Figure 1 sensors-25-01732-f001:**
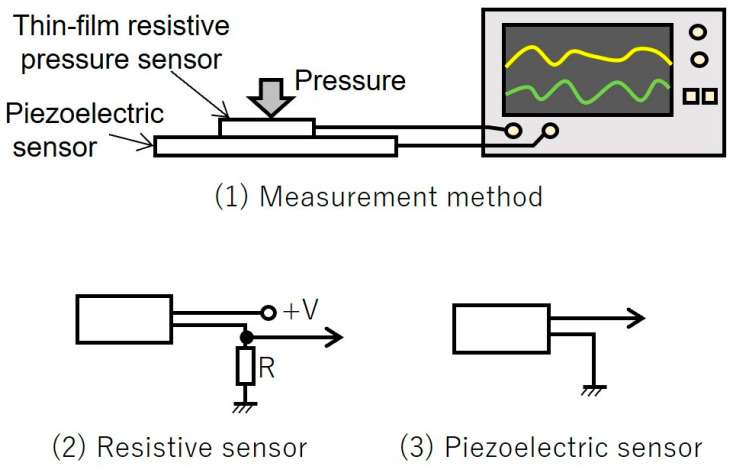
Comparison of resistive pressure sensor and piezoelectric sensor signals.

**Figure 2 sensors-25-01732-f002:**
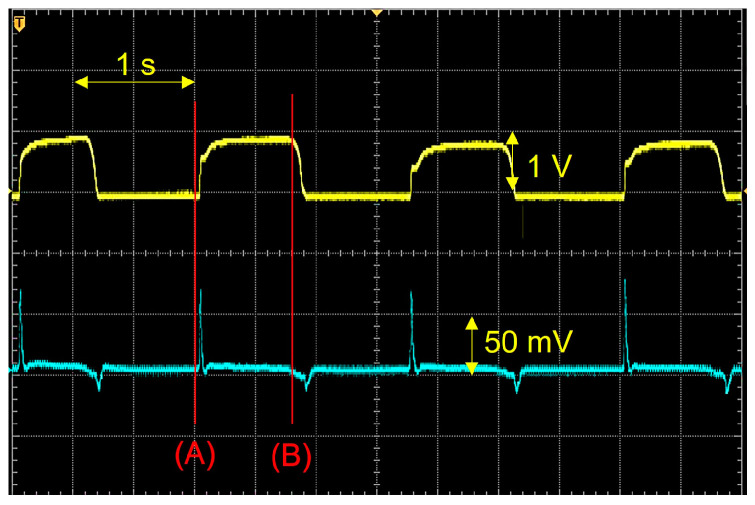
Resistive pressure sensor (**top**) and piezoelectric sensor (**bottom**) signals. (A) denotes the time of pressure increase, while (B) denotes the time of pressure decrease.

**Figure 3 sensors-25-01732-f003:**
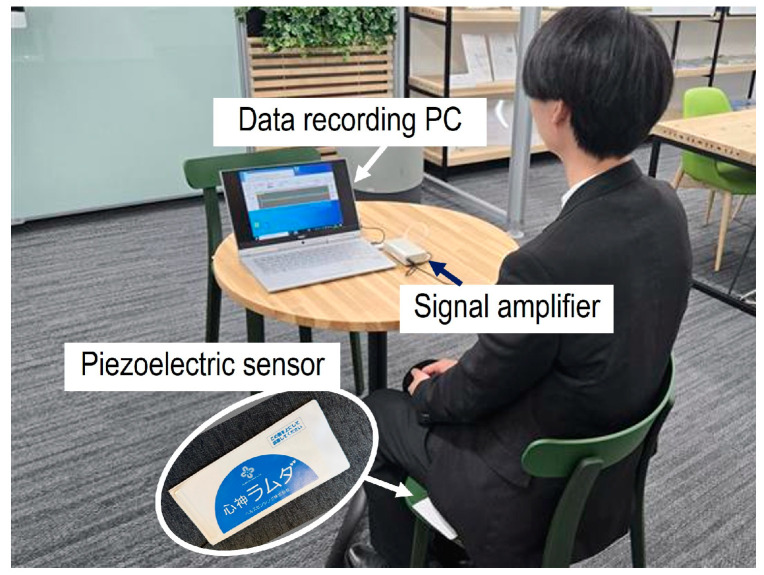
Data acquisition experiment configuration.

**Figure 4 sensors-25-01732-f004:**
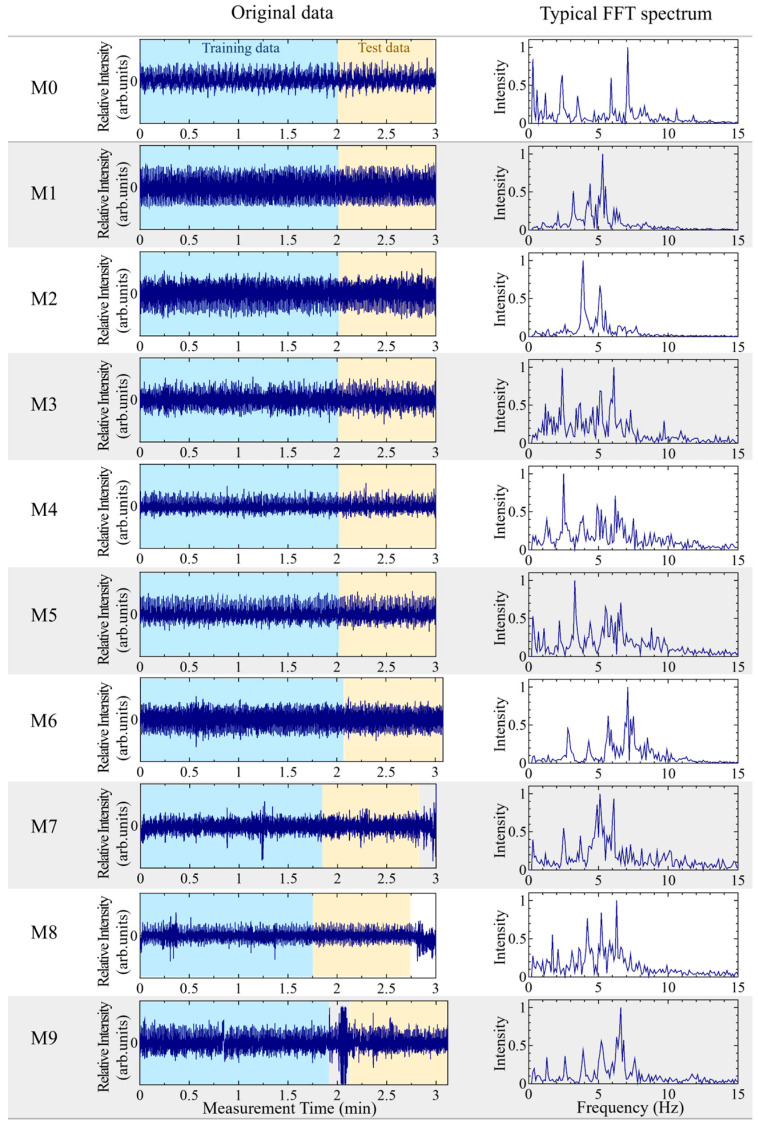
Original ballistocardial signals for M0 to M9 (**left**) and typical FFT spectrum of training data that correspond to each other (**right**). Training data are represented by the light blue shaded areas, and the test data used are represented by the yellow shaded areas.

**Figure 5 sensors-25-01732-f005:**
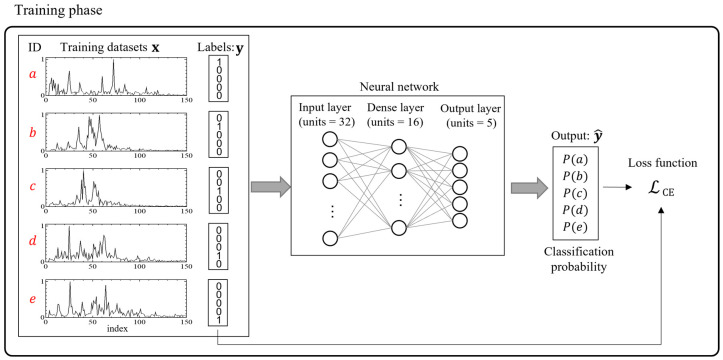
Overview of the training phases for the neural network. CE: cross-entropy. a–e represent subject IDs.

**Figure 6 sensors-25-01732-f006:**
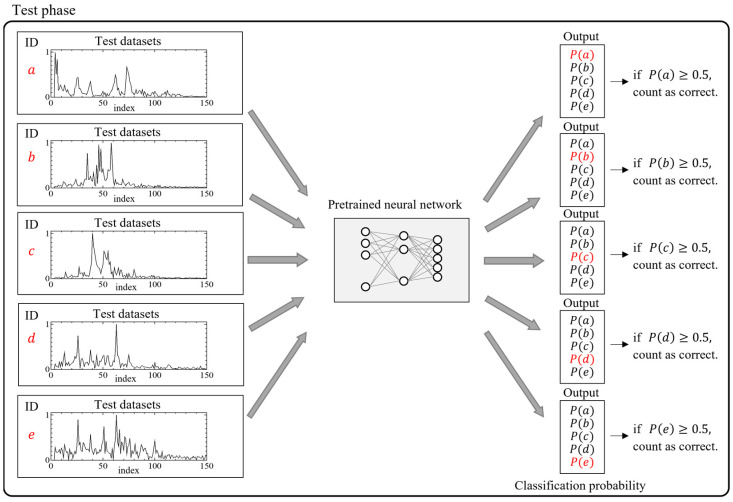
Overview of the test phases for the neural network. a–e represent subject IDs.

**Figure 7 sensors-25-01732-f007:**
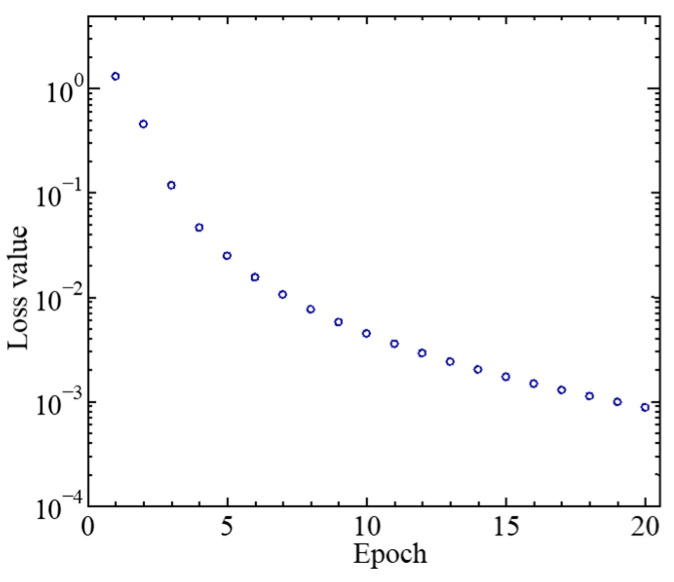
The typical loss value variation for each epoch.

**Figure 8 sensors-25-01732-f008:**
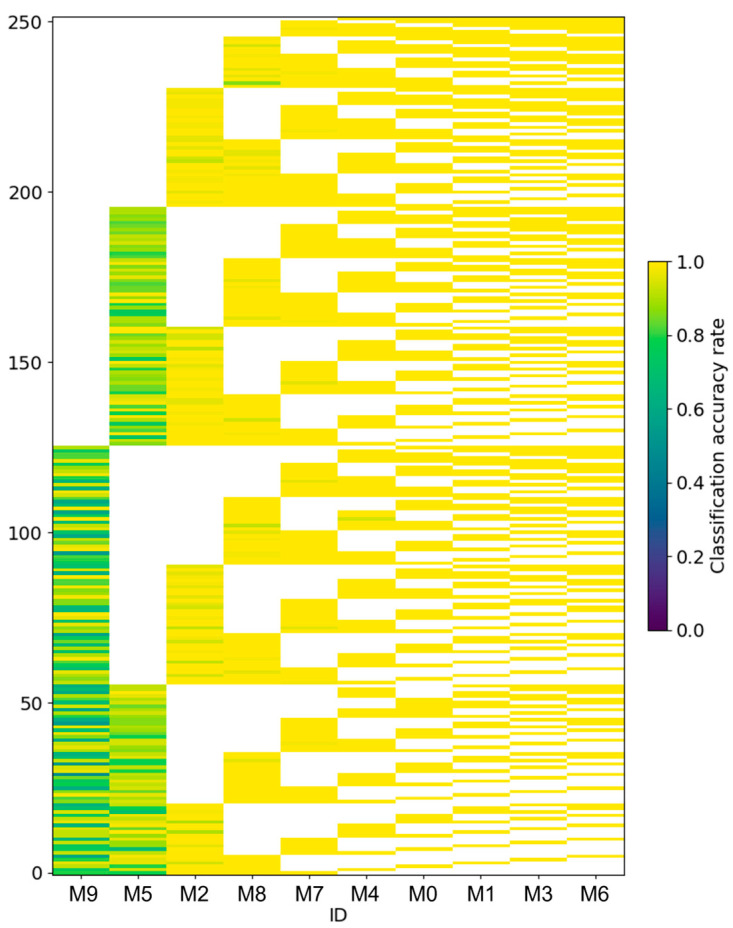
The classification accuracy rate for all 252 patterns. The horizontal axis represents the IDs, arranged in ascending order in terms of their mean classification accuracy across all patterns. The ID with the lowest mean classification accuracy is positioned on the left, whereas the ID with the highest mean accuracy is positioned on the right.

**Figure 9 sensors-25-01732-f009:**
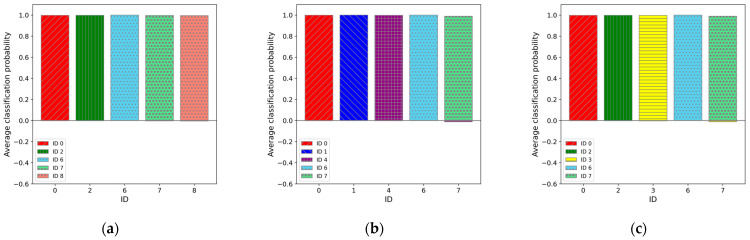
Average classification probability for 50 sets of test data in regard to each category. In regard to the classification probabilities, cases where the test data were correctly classified into its actual category were counted as positive, while cases misclassified into other categories were counted as negative. The graphs: (**a**) for IDs M0, M2, M6, M7, and M8 (“02678”) cases (best case); (**b**) for IDs M0, M1, M4, M6, and M7 (“01467”) cases (second best case); and (**c**) for IDs M0, M2, M3, M6, M7 (“02367”) cases (third best case).

**Figure 10 sensors-25-01732-f010:**
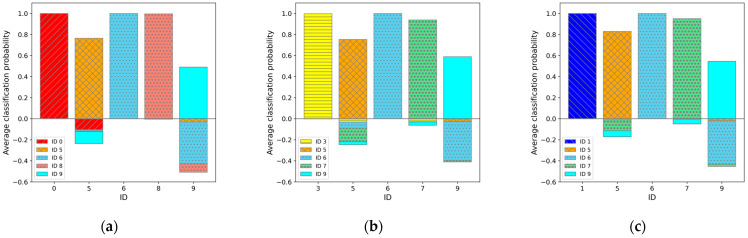
Average classification probability for 50 sets of test data in regard to each category. In regard to the classification probabilities, cases where the test data were correctly classified into its actual category were counted as positive, while cases misclassified into other categories were counted as negative. The graphs: (**a**) for IDs M0, M5, M6, M8, and M9 (“05689”) cases (worst case); (**b**) for IDs M3, M5, M6, M7, and M9 (“35679”) cases (second worst case); and (**c**) for IDs M1, M5, M6, M7, M9 (“15679”) cases (third worst case).

**Figure 11 sensors-25-01732-f011:**
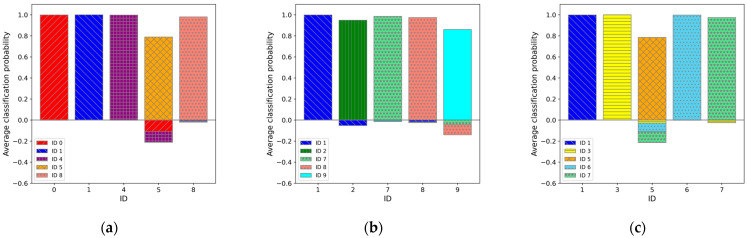
Average classification probability for 50 sets of test data in regard to each category. In regard to the classification probabilities, cases where the test data were correctly classified into its actual category were counted as positive, while cases misclassified into other categories were counted as negative. The graphs: (**a**) for IDs M0, M1, M4, M5, and M8 (“01458”) cases (average case); (**b**) for IDs M1, M2, M7, M8, and M9 (“12789”) cases (second average case); and (**c**) for IDs M1, M3, M5, M6, and M7 (“13567”) cases (third average case).

**Figure 12 sensors-25-01732-f012:**
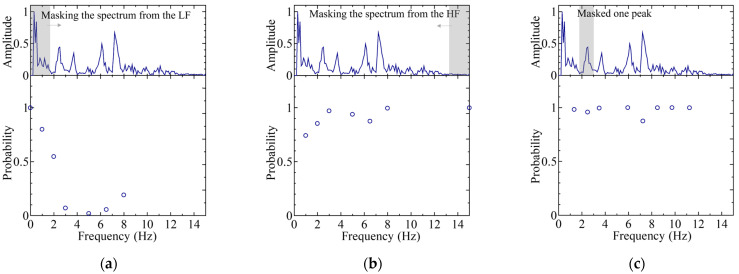
Classification probabilities for re-inference (lower graph) and a typical spectrum (upper graph) for ID M0 as part of the “02367” classification case: (**a**) masking from the low-frequency (LF) side; (**b**) masking from the high-frequency (HF) side; and (**c**) masking a single specific peak. The arrow indicates the expansion of the spectral mask.

**Figure 13 sensors-25-01732-f013:**
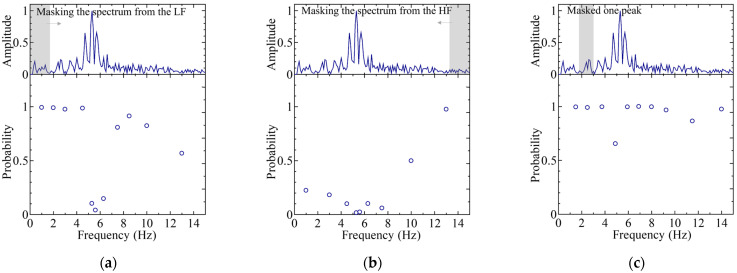
Classification probabilities for re-inference (lower graph) and a typical spectrum (upper graph) for ID M7 as part of the “02367” classification case: (**a**) masking from the low-frequency (LF) side; (**b**) masking from the high-frequency (HF) side; and (**c**) masking a single specific peak. The arrow indicates the expansion of the spectral mask.

**Figure 14 sensors-25-01732-f014:**
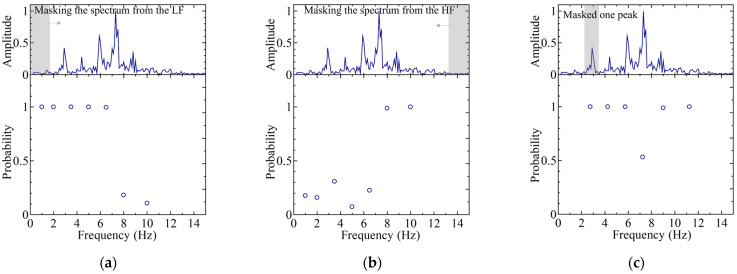
Classification probabilities for re-inference (lower graph) and typical spectrum (upper graph) for ID M6 as part of the “05689” classification case: (**a**) masking from the low-frequency (LF) side; (**b**) masking from the high-frequency (HF) side; and (**c**) masking a single specific peak. The arrow indicates the expansion of the spectral mask.

**Figure 15 sensors-25-01732-f015:**
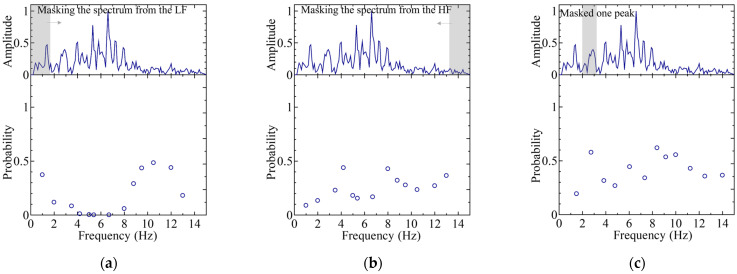
Classification probabilities for re-inference (lower graph) and a typical spectrum (upper graph) for ID M9 as part of the “05689” classification case: (**a**) masking from the low-frequency (LF) side; (**b**) masking from the high-frequency (HF) side; and (**c**) masking a single specific peak. The arrow indicates the expansion of the spectral mask.

**Figure 16 sensors-25-01732-f016:**
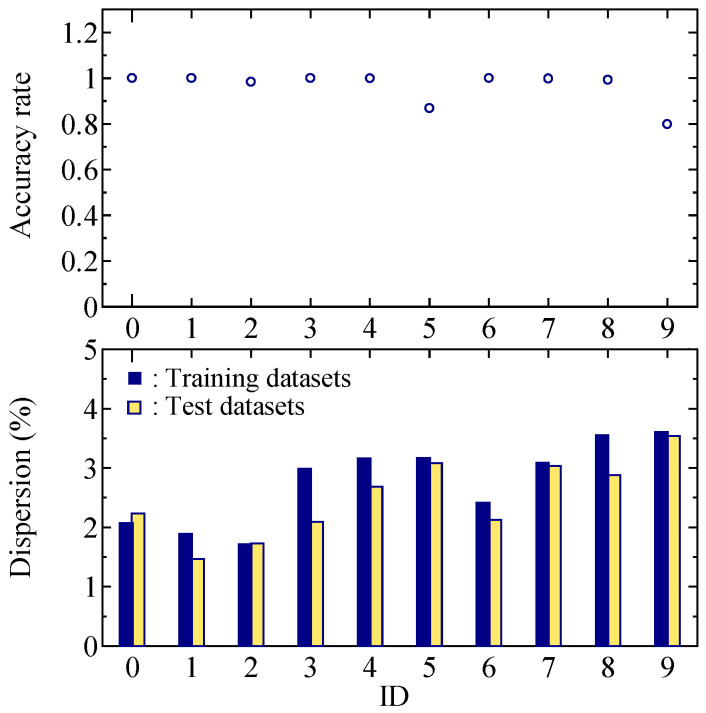
Mean accuracy rate and dispersion of the test dataset for each ID.

## Data Availability

The data supporting the findings of this study are available upon request from the corresponding author.

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
