# Peer review of "Identification of People in a Household Using Ballistocardiography Signals Through Deep Learning"

_sensors, 2025, doi:10.3390/s25061732_

Round 1

Reviewer 1 Report

Comments and Suggestions for Authors

The authors of this article present a non-invasive health monitoring system that utilizes a polyvinylidene difluoride (PVDF) piezoelectric sensor to track heart and respiratory rates without making skin contact. This approach differs from traditional sensors, which require attachment to the skin, allowing elderly individuals to monitor their health seamlessly in their daily lives.

The current study focuses on a personal identification feature based on the frequency characteristics of the biological signals collected. To achieve this, a neural network was trained and tested using data from ten subjects. Despite the limited data set, the system demonstrated high accuracy in identifying five individuals. Although the results indicate that the system is effective, future research needs to address daily variations in the signals.

I will now provide observations related to the work, addressing each chapter separately.

  • Abbreviations are typically used only after their meanings have been explained, and their use is limited in the abstract. In this case, the abbreviation for polyvinylidene difluoride appears in the abstract without being defined beforehand. It may be better to first present the full term and introduce the abbreviation in the introductory chapter.
  • The first chapter, which serves as an introduction, presents several distinct concepts. However, the connection between these concepts and the overall theme of the work is not very clear. It would be helpful to clarify the relationships among elderly individuals who can live independently or as a couple, the monitoring of their health, the presence of five individuals in a household, and the process of identifying these individuals. When conducting remote health monitoring for an elderly person, it is crucial to accurately associate registration data with the correct individual, especially if they live with others. Proper identification is essential in these situations. However, the assumption that they live alone or with a partner does not align with the five-person method used in the assessment.
  • It has been noted that the measured signals (M7-M9) include body movement (lines 149-149). Was this inclusion intentional or accidental? Earlier, you mentioned that the acquisition data would be longer in the presence of disturbances. Could you please clarify this further?
  • Why were M0 and M6 selected as the best-case examples? While there is a clear rationale for choosing the signals in the worst-case scenario (M9 is often misclassified as M6), the reasoning for the best-case selection is not as apparent. Could you justify the choice of examples in the favorable case as well?

Reviewer 2 Report

Comments and Suggestions for Authors

Ballistocardiogram (BCG) is a periodic weak vibration pattern of the human body caused by cardiac pulsation. By recording BCG signals, non-contact heart activity detection can be achieved. Personal identification combining BCG and deep learning is realized in the present work. 

1. experimental setup illustrated in Fig. 1:The pressure and piezoelectric sensors are directly stacked one above the other. This may result in the interference of piezoelectric sensor, since piezoelectric materials are very sensitive, such as PZT. They should be tested separately with same applied pressure.
2. If the equations 1-2, used in the present work, are not inferred by the author(s), the references should be cited.
3. BCG signal is sensitive to the movement of human body. The signals will be different under meditation and movement (turning over, swaying….). Is this considered in the deep learning research? 
4. Deep learning needs very large number of data to ensure its accuracy. Only five subjects and 252 cases are not enough.
5. Only schematical images are presented in the work. However, the image of the equipment containing sensors, signal amplifier, data recording devices should be added.
6. The details of the experiments are not explained appropriately, or the readers cannot repeat the work present in this paper.

Reviewer 3 Report

Comments and Suggestions for Authors

This manuscript investigates a non-invasive PVDF piezoelectric sensor for personal identification based on BCG signals. The study demonstrates promising results in classifying individuals by analyzing frequency components of BCG signals using a neural network (NN). The approach shows potential for non-contact health monitoring and personal identification within household environments. However, several aspects of the methodology and analysis need to be addressed to enhance the paper's clarity and scientific rigor. I recommend major revisions before considering the manuscript for publication.
1.    The study uses a PVDF piezoelectric sensor to capture BCG signals. What specific advantages does PVDF provide over other sensor materials in this application?
2.    The data are collected by participants sitting on a sensor for around 3 minutes. Could the results be affected by any changes in posture, movement, or even slight variations in seating conditions? How consistent are the sensor signals across different seating positions?
3.    In the worst classification cases, body movement is highlighted as a factor contributing to misclassification. How is body movement quantified, and is there a specific threshold for what constitutes “significant” movement during data collection? Although there is only one case, the classification accuracy of less than 50% is too low. How can it be improved?
4.    The paper mentions that the main peak of the frequency spectrum is a key feature for classification. Could you clarify how this main peak is identified in each individual’s signal?
5.    The neural network (NN) architecture uses a simple feedforward model with ReLU activations. Why not use more complex neural network architectures (e.g., convolutional networks) to capture temporal or spatial features in signals?
6.    The study uses data from 10 participants. How representative is this dataset of the broader population, particularly for elderly individuals with varying physical conditions or mobility levels?
7.    The NN model performs well with a training set of 500 samples per individual. How did the authors ensure that the training data was not overfitted? Cross-validation experiments are recommended.
8.    The manuscript contains grammar and spelling issues that need careful revision. For example, the phrase “Signals obtained when approximately the same pressure were applied to…” should be corrected to “Signals obtained when approximately the same pressure was applied to…” The authors should thoroughly proofread the text to improve its language quality.

Comments on the Quality of English Language

The manuscript contains grammar and spelling issues that need careful revision. 

Round 2

Reviewer 2 Report

Comments and Suggestions for Authors

Author(s) addressed my questions in the present revised manuscript. There is no further issues.

Reviewer 3 Report

Comments and Suggestions for Authors

The authors have already revised the manuscript according to the comments.